# A cross-circulatory platform for monitoring innate allo-responses in lung grafts

**Matthieu Glorion[1]☯, Florentina Pascale[1]☯, Jérôme Estephan[2], Maxime Huriet[2], Carla Gouin[2], Céline Urien[2], Fany Blanc[3], Julie Rivière[3,4], Christophe Richard[5], Valérie Gelin[5], Julien De Wolf[1], Morgan Le Guen[6], Antoine Magnan[7], Antoine Roux[7], Isabelle Schwartz-Cornil [2]*, Edouard Sage[1,2]**

**1** Department of Thoracic Surgery and Lung Transplantation, Foch Hospital, Suresnes, France, **2** Université Paris-Saclay, INRAE, UVSQ, VIM, Jouy-en-Josas, France, **3** Université Paris-Saclay, INRAE, AgroParisTech, GABI, Jouy-en-Josas, France, **4** Université Paris-Saclay, INRAE, AgroParisTech, MICALIS Institute, Jouy-en-Josas, France, **5** Université Paris-Saclay, INRAE, UVSQ, BREED, Jouy-en-Josas, France, **6** Department of Anesthesiology, Foch Hospital, Suresnes, France, **7** Department of Pulmonology, Foch Hospital, Suresnes, France

☯ These authors contributed equally to this work.
* isabelle.schwartz@inrae.fr

**Data Availability Statement:** All relevant data are within the paper and its Supporting Information files.

## Abstract

Lung transplantation is the only curative option for end-stage chronic respiratory diseases. However the survival rate is only about 50% at 5 years. Although experimental evidences have shown that innate allo-responses impact on the clinical outcome, the knowledge of the involved mechanisms involved is limited. We established a cross-circulatory platform to monitor the early recruitment and activation of immune cells in an extracorporeal donor lung by coupling blood perfusion to cell mapping with a fluorescent marker in the pig, a commonly-used species for lung transplantation. The perfusing pig cells were easily detectable in lung cell suspensions, in broncho-alveolar lavages and in different areas of lung sections, indicating infiltration of the organ. Myeloid cells (granulocytes and monocytic cells) were the dominant recruited subsets. Between 6 and 10 h of perfusion, recruited monocytic cells presented a strong upregulation of MHC class II and CD80/86 expression, whereas alveolar macrophages and donor monocytic cells showed no significant modulation of expression. This cross-circulation model allowed us to monitor the initial encounter between perfusing cells and the lung graft, in an easy, rapid, and controllable manner, to generate robust information on innate response and test targeted therapies for improvement of lung transplantation outcome.

## Introduction

Allogeneic lung transplantation (LT) is the sole therapeutic option for terminal chronic respiratory diseases. While LT practice is increasing worldwide, the outcomes are disappointing with a median survival of 6 years [1]. The major non-infectious early complication of successful surgery is severe primary graft dysfunction (PGD), a non-cardiogenic pulmonary edema syndrome that occurs in the first 3 days post LT, in about 11–25% of patients [2]. In most

**Funding:** ES received funding from Association Chirurgicale Pour Le Développement et L'Amélioration des Techniques de Dépistage et de Traitement des Maladies Cardio-vasculaires (ADETEC-Coeur) and from la « Chaire Universitaire de Transplantation Université de Versailles-Saint-Quentin en Yvelines, Hôpital Foch et Fondation Foch, AR received a grant from the Association Gregory Lemarchal and the association Vaincre la Mucoviscidose (project number RF20220503016) and ISC received funding from INRAE institutional support. The funders had no role in study design, data collection and analysis, decision to publish, or preparation of the manuscript.

**Competing interests:** The authors have declared that no competing interests exist.

**Abbreviations:** Lung transplantation, LT; alveolar macrophages, AMs; monocytic cells, MoCs; dendritic cells, DCs; polymorphonuclear neutrophils, PMNs; primary graft dysfunction, PGD; carboxyfluorescein succinimidyl ester, CFSE; pulmonary artery, PA; pulmonary vein, PV; swine, sw; human, hu; broncho-alveolar lavage, BAL; isotype control, ISC; single cell RNA-seq, scRNA-seq; hematoxylin-eosin-saffron, HES.

patients, chronic lung allograft dysfunction (CLAD) develops following a series of rejection and reparation events and stands as a major impediment to long term survival [3].

Several experimental works support that the innate allogeneic response occurring during surgery strongly impacts on the clinical outcome of LT, i.e. PGD occurrence, rejection events and CLAD development [4, 5]. At the surgical step, the brutal reoxygenation of the hypoxic graft upon de-clamping, so called ischemia-reperfusion, generates a huge production of reactive oxygen species, inflammatory cytokines and Damage-Associated Molecular Pattern molecules, leading to ischemia-reperfusion injuries [5]. Additional activation signaling implicates the allo-recognition through innate molecules such as CD47/CD172A and paired immunoglobulin-like receptors (PIR)/MHC class I, as shown in mouse models [6, 7]. The innate responses of alveolar macrophages (AMs), polymorphonuclear (PMNs) cells and monocytes have been shown to be instrumental in the PGD pathogenesis in mouse models of allogeneic LT [8, 9]. The combination of inflammation and allo-recognition is thought to trigger the differentiation of monocytes into inflammatory dendritic cells (DCs) that will then stimulate allogeneic T and B cell responses leading to chronic rejection [5, 6]. Knowledge of the different cell types and mechanisms involved in the allogeneic innate response of LT is fragmentary and needs to be tackled to design suitable therapeutics, in preclinical models of better translational relevance than mice.

Pig is a highly pertinent model for lung transplantation and is commonly used in translational research in this field [10]. In order to get more insight into the innate allogeneic response in LT using this species, we developed a model inspired from a cross-circulation platform that was established by a New York team to repair damaged lungs [11, 12] which was shown to be amenable to a 4 day-duration [13]. On this platform, an extracorporeal donor lung is connected to the blood circulation of the perfusing pig, and we coupled to it to a cell mapping strategy by a systemic CFSE injection into the perfusing pig. Compared to full transplantation, this relatively simple, rapid and highly controllable model enables repeated sampling of extracorporeal lung fragments to study the cell recruitment and activation. We show here that the innate allo-immune response involves the recruitment of many cell types with the dominant implication of myeloid cells, i.e. PMN and monocytic cells (MoCs). Furthermore the expression of MHC class II and CD80/86 was systematically enhanced on recruited MoCs between 6 h and 10 h and not significantly on donor lung MoCs and AMs. Therefore the cross-circulation platform coupled to cell mapping allowed us to monitor the cell responses of recruited cells upon innate allo-stimulation and could be used to assess the effect of anti-inflammatory and immunomodulatory drugs.

## Material and methods

### Ethics approval

The animal experiments were conducted in accordance with the EU guidelines and the French regulations (DIRECTIVE 2010/63/EU, 2010; Code rural, 2018; Décret n°2013–118, 2013). The experiments were approved by the COMETHEA ethics committee under the APAFIS number authorization 25174–2020011414322379 and were authorized by the French "ministère de l'enseignement supérieur et de la recherche". The authors complied with the ARRIVE guidelines.

### Animals

Twenty Large-White pigs (ten donor–recipient pairs) were hosted in the Animal Genetics and Integrative Biology unit (GABI-INRAE, France). Matched pairs of male donors and female

recipients from different siblings were used. The animals were 3–5 months of age, with a mean weight of 50.8 ± 4.4 kg.

## Donor lung harvest and lung cannulation

The lung harvest from donor pigs (*n* = 10) was performed in the Animal Surgery and Medical Imaging Platform (CIMA-MIMA2-BREED-INRAE, Jouy en Josas, France). Heart-lung mono-bloc harvests were performed using a non-heart-beating donor swine model [14]. Anesthesia was induced by a combination of 1 mg/kg Rompun® 2% (Elanco, Heinz-Lohmann-Strasse 4, Cuxhaven,Germany) and 15 mg/kg Imalgene® 1000 (Boehringer Ingelheim Animal Health, Lyon, France) and followed by 6% isofluorane. A 25,000 U heparin bolus (Sanofi, Paris, France) was administered i.v.. Pigs were euthanized with 50 mg/kg Dolethal® (Vetoquinol, Magny-Vernois, France). After cardiorespiratory arrest, a period of 10 min of no touch was applied. After median sternotomy, a cannula was placed and secured in the main pulmonary artery (PA). The lungs were flushed with 3 L of Perfadex® (XVIVO Perfusion, Göteborg, Sweden) administered in a cold anterograde perfusion. The heart-lung block was explanted with inflated lungs to a sustained airway pressure of 15 cmH2O, and the trachea was stapled (Endo GIA device, Medtronic, Dublin, Ireland). The block was placed in a cooling jacket for the dissection. The heart was removed, leaving behind a circumferential left atrial cuff. Dedicated cannula (XVIVO Perfusion) were sutured to the left atrium with 5–0 Prolene® (Ethicon, Somerville, NJ, USA). The arterial cannula was fixed to the PA with a purse string of Mersutures® 1 (Ethicon). A cold retrograde flush with 1 L Perfadex® (XVIVO Perfusion) was performed. The lungs were placed in a sterile isolation bag with 500 ml Perfadex® and stored at 4˚C in the refrigerator. The mean duration of warm ischemia between euthanasia and cold storage was 82.8 ± 9.3 min. The mean duration of cold ischemia between cold storage and cross circulation initiation was 103.9 ± 15.6 min.

## Perfusing pig conditioning

Pigs (*n* = 10) were anesthetized as described above. Septotryl® (0.08 ml/kg) (Vetoquinol) was injected i.m. prior to catheterization. A femoral arterial line (Arrow International, Cleveland, Ohio, USA) was placed percutaneously under ultrasound guiding for hemodynamic monitoring (S1 Fig). A 25,000 U heparin bolus was administered. The superior vena cava was cannulated with a 20 F double lumen cannula (Avalon Elite, Maquet Cardiopulmonary, Rastatt, Germany) using the Seldinger percutaneous technique [15]. The correct positioning of the wire and cannula in the inferior vena cava were checked by ultrasound (S1 Fig). The pig was positioned on the left lateral decubitus with a heating blanket to avoid hypothermia. Physiological parameters, including heart rate, electrocardiogram, blood pressure, oxygen saturation, end-tidal CO2, temperature, and respiratory rate were continuously monitored. About 30 min before the initiation of cross-circulation, 25 mg carboxyfluorescein succinimidyl ester (CFSE) (Sigma-Aldrich, Saint-Louis, MS, USA) was administered i.v. into the perfusing pig, diluted in 4 ml DMSO + 40 µl heparin. Sedation was maintained for 10 h by continuous administration of 2–4 mg/kg/h propofol (Proposure®, Axience, Pantin, France) + 0.6% isofluorane and analgesia was obtained by administration of 0.2 mg/kg nalbuphine i.v. every 3 hours.

## Cross-circulation

We basically followed the procedure described in [12, 15]. The circuit was filled with 1.5 L ringer lactate (Vetivex®, Dechra Northwich, UK). As shown in Fig 1, the circuit consists in a main console (SCPC centrifugal pump console, LivaNova, London, UK), a disposable pump (Revolution centrifugal pump, LivaNova), a hard-shell reservoir (LivaNova) and three 8-inch

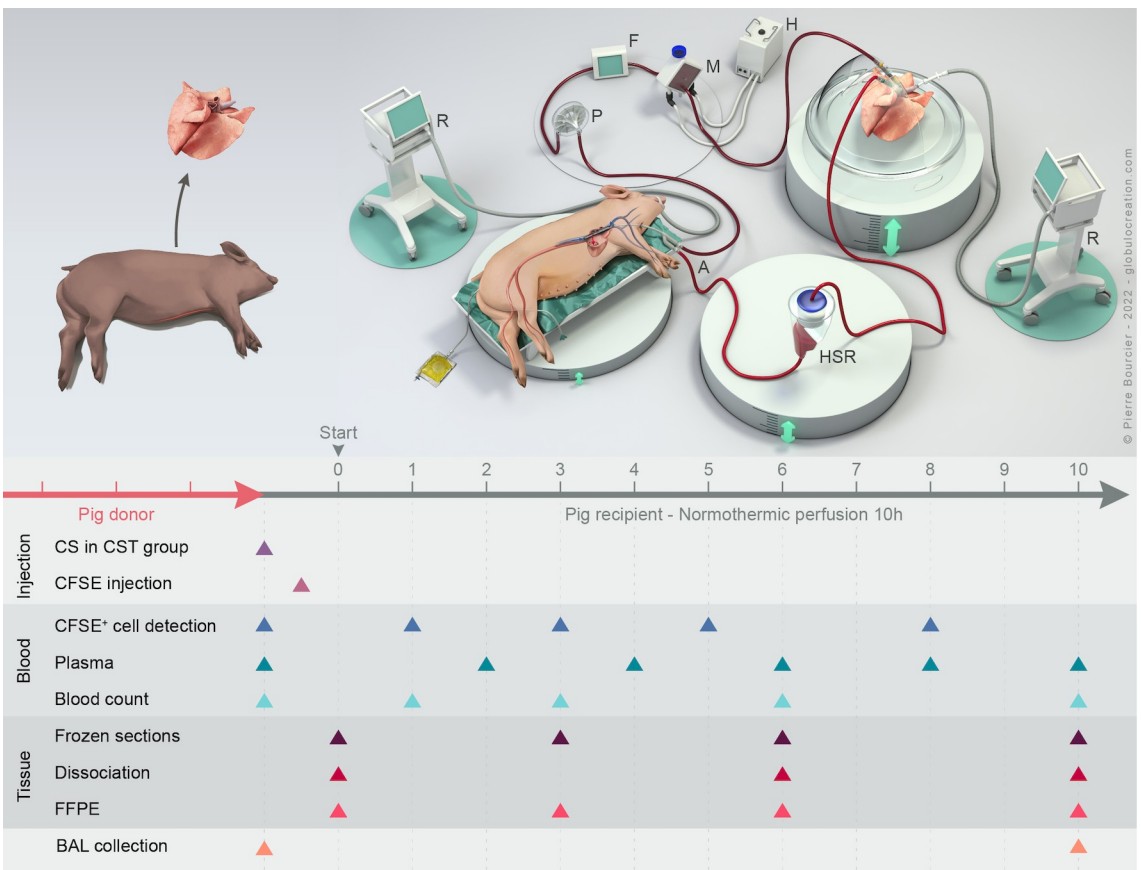

**Fig 1. Experimental overview of the cross-circulation model for cell mapping. Top.** Setup of the cross-circulation technique. A double lumen canula, HSR hard-shell reservoir, R ventilator, H heater, M membrane, F flowmeter, P pump. **Bottom.** Experimental timeline of injection procedures and sample collections. CFSE (25 mg) was injected 30 min before cross-circulation. Blood, broncho-alveolar lavage (BAL) and lung tissue biopsies were collected at different time points. Blood was used for the detection of CFSE+ cells in the perfusing pig, for biochemical profiling in plasma, and for blood cell counts. Lung tissue was used to produce frozen sections (location of CFSE+ cells in the tissue), for enzymatic dissociation (CFSE+ cells composition in the tissue), and for formalin-fixed paraffin-embedded section generation (histological analyses). BAL was used for CFSE+ cell detection.

tubing (Smart coated tubing, LivaNova). An EOS® oxygenator (LivaNova) was used to heat the circuit. The gas connection was occluded. The PA and pulmonary vein (PV) pressures, the PA flow, and the temperature data were continuously monitored. The perfusing pig was maintained on a continuous heparin infusion (100 U/Kg/h). The activated clotting time was measured using a IStat® kit (Abbott, Chicago, IL, USA) and the heparin drip was adjusted to maintain a target clotting time value of 150–200 sec. The donor lungs were placed in dorsal position on an XVIVO® chambers (XVIVO Perfusion) and the trachea was cannulated with a 7.5 mm diameter cuffed endotracheal tube (Mallinckrodt, Staines-upon-Thames, UK). The tubing was spliced to connect the perfusing pig to the dedicated circuit, marking the start of cross-circulation. Initial flow rates were set to 5% of the estimated cardiac output and were gradually increased to 10% with an initial PA target pressure below 15 mm Hg and a PV pressure of 3–5 mm Hg. Ventilation (Elisée 350 Resmed San Diego, USA) was initiated within the first 10 min with the following initial settings: volume control mode, 10/min respiratory rate, 300 mL tidal volume, 5 cm H20 positive end-expiratory pressure, and 21% FiO2. Atelectic lung regions were recruited by increasing the tidal volume and the positive end-expiratory pressure and by performing inspiratory hold maneuvers (up to 25 cm H2O). PV was

dependent on the hydrostatic pressure difference between the lungs and reservoir. Reservoir blood level was dependent on the hydrostatic pressure difference between the reservoir and the swine recipient. These two values were controlled by adjusting the height difference between the lung, the reservoir and the swine host [13].

## Extracorporeal hemodynamic and lung function monitoring

Blood samples were collected from the main PA and PV cannula every 1 h, and hemo-gas analysis was performed using a Istat® kit. Static compliance (Tidal Volume/(plate pressure–positive end expiratory pressure)), delta PCO2 (PA PCO2 –PV PCO2) and delta PO2/FIO2 ((PV PO2 –PA PO2) / FIO2). Transpulmonary pressure (PA—PV) and pulmonary vascular resistance were calculated ((PA pressure–left atrial pressure) x 80 / flow rate).

## Blood counts and biochemical monitoring

Blood samples were collected by venipuncture of the auricular vein or directly from the extracorporeal circuit after cross-circulation. Plasma was analyzed immediately for biochemical profiling. Blood count and biochemical profiling were performed on a MS4.5 analyzer and a M-Scan II analyzer (Melet Schloesing Laboratoires, Cergy-Pontoise, France).

## BAL and lung biopsies

BAL was performed before cross-circulation in the subsegmental bronchi of the azygos lobe (0 h) and after 10 h cross-circulation in the cranial and caudal lobes. Lung biopsies for cell dissociation (about 2 g) were sampled in the cranial and caudal lobes using a surgical stapler (Endo GIA™ universal stapling system, Medtronic, Minneapolis, USA) and were immediately immerged in cold hypothermic preservation media (HypoThermosol® FRS, Stemcell Technologies Inc, Vancouver, Canada). For immuno-histo-fluorescence, biopsies (about 5 mm³) were snap-frozen in a matrix gel (Sakura, Paris, France). For histology, biopsies were fixed in cold phosphate-buffered 4% paraformaldehyde for 24 h and subsequently paraffin-embedded.

## Lung cell extractions

Tissues (2 g) were minced and incubated for 45minutes at 37˚C on a rotary shaker in RPMI 1640 supplemented with 100 IU/ml penicillin,100 μg/ml streptomycin, 2 mM L-glutamine and 10% inactivated fetal calf serum (FCS) (all from Invitrogen, Paisley, UK), containing 3 mg/ml collagenase D, 0.25 mg/ml Dnase I (Sigma-Aldrich) and 0.7 mg/ml dispase II (Gibco®, ThermoFisher Scientific, St Aubin, France). The minced preparation was crushed and filtered on a nylon mesh (1 mm diameter) and filtered through successive cell strainers (500 μm, 100 μm, 40 μm). Red blood cells were lysed with erythrocytes lysis buffer (10 mM NaHCO3, 155 mM NH4Cl, and 10 mM EDTA). After a wash in PBS, $10^8$ cells were used for cell surface staining. The rest of the cells was frozen in FCS + 10% DMSO in a Mister Frosty freezing container (Nalgene, Rochester, NY, USA) and kept in liquid N2.

## Cell surface staining and flow cytometry

Cell surface staining was performed in RPMI + 10 mM Hepes supplemented with 5% horse serum and 5% swine serum (Gibco, Life Technologies Europe, Bleiswijk, Netherlands). Primary and secondary Abs and their working dilutions are listed in S1 Table. Matched isotype controls for mouse IgG1, IgG2b, and IgG2a were used at the same concentration as the corresponding mAbs of interest, using the fluorescence minus one method [16]. In cases of third step labelling with a directly conjugated mAb of the same IgG1 isotype as the primary Ab

(anti-CD163-RPE and anti-CD3-RPE), an excess of mouse IgG1 (50 μg/ml) was used in an additional saturation step. Dead cells were excluded by DAPI staining (Sigma-Aldrich). Samples were acquired on BD LSR Fortessa™ Cell Analyzer (BD-Biosciences). Acquired data were analyzed using FlowJo software (version 10.7.1; Tree Star, Ashland, OR, USA).

### Histology

Formalin-fixed paraffin-embedded lung tissues at 0, 6, 10 h were sectioned every 50 μm for generating six 5 μm tissue slices per sample and stained with hematoxylin-eosin-saffron (HES). The slides were imaged with a slide scanner (Pannoramic SCAN II, v3.0.2, 3DHistech, Medipixel Ltd, Budapest, Hungary) and analyzed by an external pathologist and a veterinarian in a blinded fashion. Five randomly selected high power fields ($7x10^4$ μm$^2$ area) from 6 slides per sample were observed and scored by quantification of airway and alveolar polymorphonuclear cells and interstitial edema according to a reference scoring [11].

### Immunohistofluorescence

Cryosections (10 μm) of lung parenchyma frozen biopsies were obtained using a cryostat (Leica CM3050S, Nanterre, France). Sections were fixed in methanol/acetone (1:1) at -20˚C for 20 min and stained with sheep IgG anti-FITC IgG in order to amplify the CFSE signal, followed by donkey anti-sheep IgG-A594 and with matched sheep IgG controls (S1 Table). Sections were stained with DAPI and mounted in SlowFade medium (Invitrogen). The sections were scanned at a x 20 magnification with the Pannoramic SCAN II, v3.0.2.

### Statistics

Data were analyzed with the GraphPad Prism 8.0 software. After subjecting the data to a normality test, a paired parametric two-tailed t-test was used to compare values between different time points. When the data did not present a normal distribution (vital parameters), a non-parametric Wilcoxon signed rank test was used.

## Results

### 1. Extracorporeal perfusion of lungs with an allogeneic cross-circulation platform maintains lung functions and structures over 10 h

To analyze the initial donor/recipient cell encounter in lung allografts, we adapted the cross-circulation platform that has been developed recently in pigs [11, 12, 15]. To generate ischemia-reperfusion in allogeneic conditions, we used an outbred donor and a perfusing pig of different sex originating from independent families and the donor lungs were explanted upon circulatory death with about 80 min warm ischemia. The perfusion was done by cannulating the superior vena cava of the recipient and connecting the blood circulation to an extracorporeal circuit warmed at 37˚C that perfuses the donor lung, as described in Fig 1 and S1 Fig. For cell mapping purposes, we proceeded to an i.v. injection of CFSE in the perfusing pig about 30 min before initiating the perfusion in order to label most circulating leukocyte cells [17]. The static compliance, ΔPO2:FiO2 and pulmonary vascular resistance were within effective ranges and remained stable in the graft throughout the whole procedure (Fig 2A–2D). In the perfusing pigs, vital parameters (heart beats, blood pressure, creatinine, lactate, glucose levels) remained within normal values (S2 Table). Blood counts slightly varied over time within the normal range (S2 Table). Gross appearance remained normal, without signs of pulmonary edema or hemorrhage (S2 Fig). Cell viability was evaluated with exclusion of DAPI on dissociated lung cell preparations and indicated that the cross-circulatory process preserved cellular

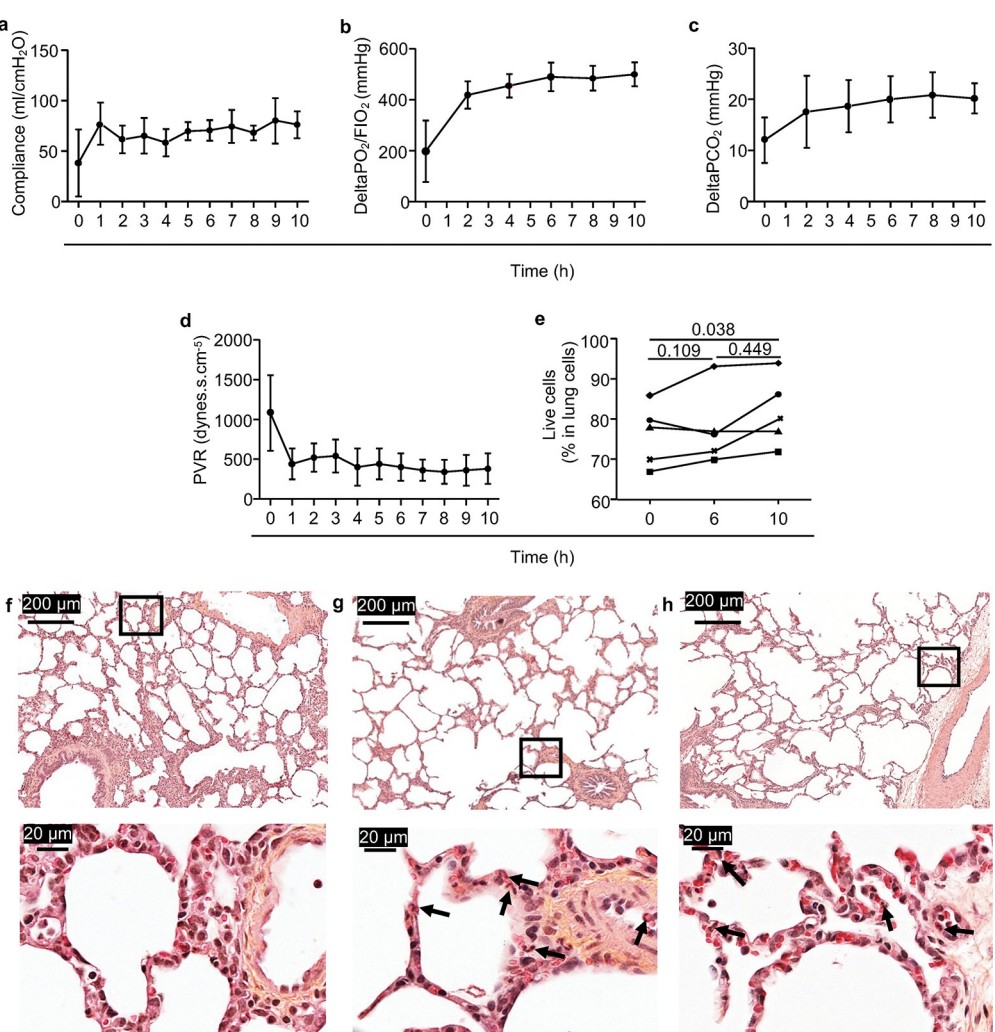

**Fig 2. Extracorporeal lung stability and performance throughout 10 h of cross-circulation.** In a, b, c, d, all values represent means ± standard deviations (n = 5) **a**. Static compliance. **b**. $\Delta PO_2/FIO_2$: (venous $PO_2$—arterial $PO_2$)/$FIO_2$. **c**. $\Delta PCO_2$ = arterial $PCO_2$—venous $PCO_2$. **d**. Pulmonary Vascular Resistance (PVR): ((pulmonary arterial pressure—left atrium pressure) x 80) / flow rate. **e**. Live cells in total lung cells evaluated with exclusion of DAPI staining from enzymatically dissociated lung fragments collected at 0, 6, 10 h. Each pig was labelled with a unique symbol throughout the paper. After passing a normal distribution test, the values were compared between different timings with a paired t-test and the p-values are reported. **f, g, h**. Conventional histopathological assessment (hematoxylin and eosin staining) of the pig lung at 0 h (f), after 6 h (g) and 10 h (h) of cross-circulation. A higher magnification of the squared field on top is shown at the bottom of each panel. See the weak infiltration with PMNs around blood vessels (black arrows, < 25 PMNs per high power field, 30 fields/sample).

viability over time (Fig 2E). Conventional blinded histopathological assessment revealed the preservation of airway and parenchymal structures over time, no sign of oedema and a weak score of infiltration with inflammatory polymorphonuclear cells (PMNs), mainly around blood vessels (Fig 2F–2H).

The ischemia-reperfusion in the allogeneic context of a cross-circulatory platform with CFSE injection induced minimal changes in the lung tissue structure, and maintained the hemodynamic, respiratory, and vital functions.

## 2. The perfusing CFSE$^+$ cells in the *ex vivo* lung during cross-circulation were recruited to various lung territories

We next analyzed the distribution of the CFSE$^+$ cells in the perfusing pig blood and in different donor lung compartments. Flow cytometry analysis of the blood cells showed that the majority of the cells ($>$ 80%) were CFSE-labelled 1 h after initiation of cross-circulation (Fig 3A). This systemic staining slowly decreased over time, with a majority of cells being still stained after 8 h (S3 Fig). Cell suspensions from enzymatically dissociated lung fragments collected at 0, 6, 10 h post-initiation of cross-circulation were analyzed by flow cytometry. The lungs had been flushed with saline at 0 and 10 h for removing most blood cells but not at 6 h, to avoid perturbing the perfusion. The CFSE$^+$ cells analyzed by this technique will be designated as recruited cells from the recipient pigs, not precluding whether the cells are located in the blood lumen and in the lung parenchyma. Fig 3B shows that the percentage of CFSE$^+$ lung cells reached 31.14% ± 8.03 at 6 h and 24.78% ± 6.31 at 10 h (no significant difference between 6 h and 10 h). The lack of increase of the CFSE$^+$ cell fraction between 6 and 10 h could be related to the flushing omitted at 6 h combined with a decrease in CFSE intensity at later timing. CFSE$^+$ cells were also found in the bronchoalveolar lavages (BAL) at 10 h, indicating that some of the recruited CFSE$^+$ cells proceeded to diapedesis through lung vascular barriers (Fig 3C). Note that the vast majority of cells in BAL cells are alveolar macrophages (AMs) which are from the donor and are therefore CFSE$^-$. The CFSE signal was sufficiently high for FACS detection but not for detection with microscopy. We used anti-FITC sheep IgG to amplify the CFSE signal (S1 Table) and we found CFSE$^+$ cells dispersed throughout the lung tissue on cryosections (Fig 3D).

Therefore, cell tracking with CFSE staining shows that recipient cells are recruited to various lung territories during the cross-circulation.

## 3. Myeloid subsets were the main cell types recruited upon cross-circulation

We proceeded to multiparameter FACS analyses of the recruited CFSE$^+$ cells in order to analyze their composition in different myeloid and lymphoid cell subsets (Fig 4, S4 Fig). The dominant subset among CFSE$^+$ cells in both groups was PMNs (44.02 ± 5.02% at 6 h and 42.92 ± 11.33% at 10 h) followed by SSC-A$^{lo}$CD172A$^{hi}$ cells (9.62 ± 2.34% at 6 h and 9.23 ± 2.38% at 10h) that are monocytic cells (MoCs) in the pig [18]. Upon their recruitment in the lung, monocytes from blood could start their differentiation into macrophages, a status which could not be appreciated with the current staining, therefore we will keep the designation of these cells as MoCs [19]. Other subsets represented less than 10% of the CFSE$^+$ cells and included SSC$^{lo}$CD172A$^{int}$ cells (not known population), CD13$^+$ cells previously shown to represent conventional dendritic cell type 1 (cDC1, [20]), NKp46$^+$ cells (natural killer), CD21$^+$ cells (mature B-cells [21]), and CD4$^+$CD3$^+$ and CD8$^+$CD3$^+$ T-cells.

The cross-circulation platform revealed that PMNs and MoCs are the main populations of recruited cells in the lung at the early 6 h and 10 h time points.

## 4. The recruited MoCs from the perfusing pig upregulate their expression of MHC class II and CD80/86, whereas donor MoCs and AMs showed no significant modulation of expression

Expression of MHC class II and CD80/86, which are major molecules driving antigen presentation, can be upregulated on monocytes upon ischemia-reperfusion [22]. In addition monocytes and AMs have been shown to play an important role in the ischemia-reperfusion response and PGD in a mouse model [9, 23] and these cells are expected to play a role in antigen-presentation leading to rejection [5, 6, 24].

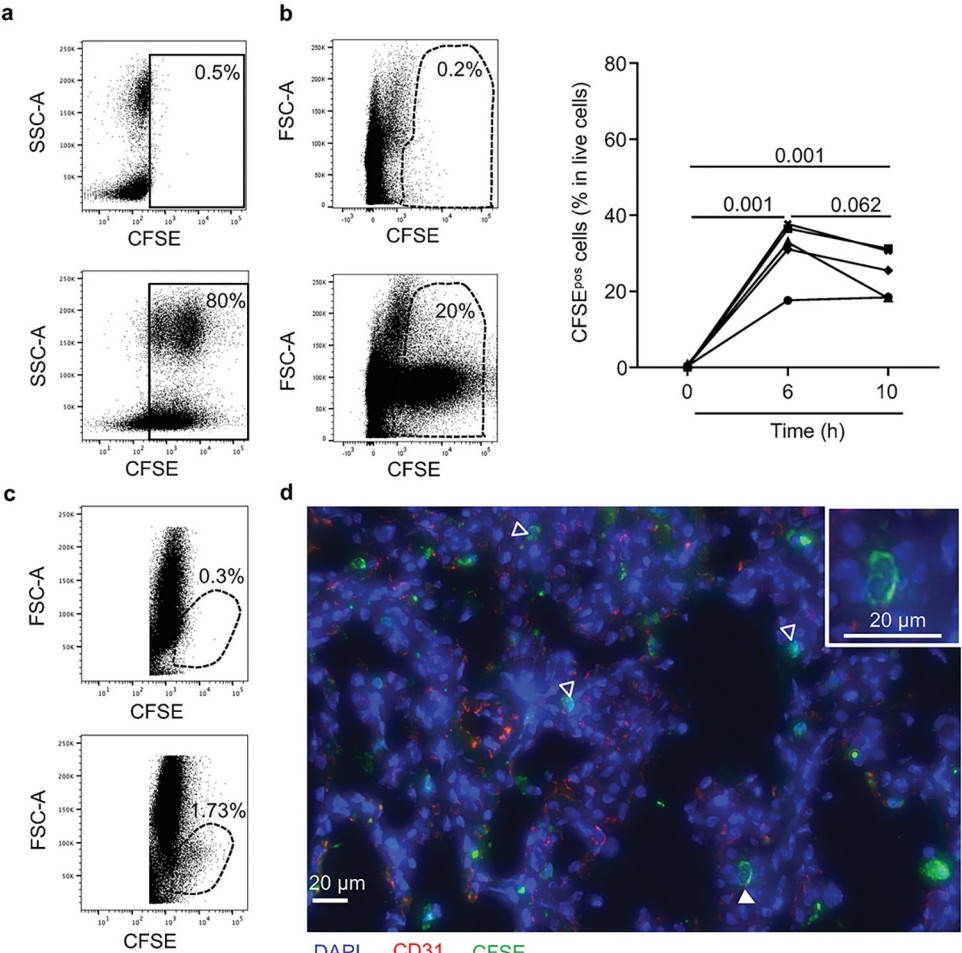

**Fig 3. Detection of CFSE+ cells in the perfusing pig blood and in the lung upon cross-circulation. a.** Blood was collected 1 h before and 1 h after the cross-circulation initiation (top and bottom respectively), and CFSE was injected 30 min. before cross-circulation initiation. Whole blood cells were analyzed by flow cytometry after lysis of erythrocytes and the percentage of CFSE+ cells is indicated. **b.** Lung biopsies were collected at 0 h, 6 h and 10 h of cross-circulation and a single cell suspension was generated by enzymatic treatment. Left, the FACS profiles at 0 h (top) and 10 h (bottom) of one representative experiment are shown and the % CFSE+ among DAPIneg live cells is indicated. Right, the proportion of CFSE+ cells among live lung cells is shown (5 pigs/group). Each pig is labelled with a unique symbol that is conserved throughout the paper. After passing a normal distribution test, the values were compared between different timings with a paired t-test and the p-values are reported. **c.** BAL was collected at 0 h from the azygos lobe (top) and after 10 h of cross-circulation from the rest of the lung (bottom), stained with DAPI and analyzed by flow cytometry. The live CFSE+ cell population is shown. **d.** Cryosections of lung biopsies were fixed with acetone:methanol, stained with APC-conjugated anti-swine CD31 (red) and sheep anti-FITC IgG followed by A594-conjugated anti-sheep IgG in order to amplify the CFSE signal for microscopy (green). Nuclei were counterstained with DAPI (blue). The sections were scanned at a x 20 magnification with the Pannoramic SCAN II. The white filled arrow points to a CFSE+ monocyte expanded at a higher magnification. White empty arrows point towards CFSE+ cells that appear located within the parenchyma and/or at the alveolar side.

In order to test whether the cross-circulation platform could reveal MoC and/or AM activation, we analyzed MHC class II and CD80/86 modulation of expression on MoCs and AMs at 6 h and 10 h in the perfused lung. When focusing on the recruited CFSE+ MoCs, we found that the percentage of MHC class II+ cells increased from 21.8 ± 7.1% at 6 h to 38.6 ± 3.04% at 10 h (p = 0.008) (Fig 5A left, and S5 Fig). Similarly, the percentage of CD80/86+ cells in CFSE+ MoCs cells increased from 11.41 ± 5% at 6 h to 24.26 ± 5% at 10 h (p = 0.002, Fig 5B left and S5 Fig).

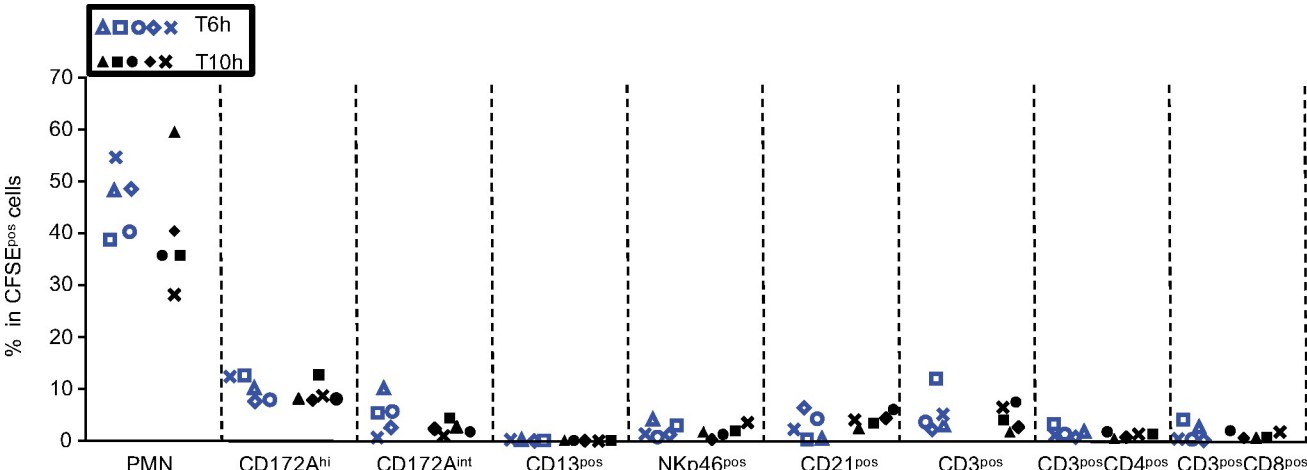

**Fig 4. Recruitment of immune cell subsets upon 6 and 10 h of cross-circulation.** Contribution of immune cell subsets in the CFSE$^+$ cells. Within live CFSE$^+$ gated cells, the percentage of PMNs, SSC-A$^{lo}$CD172A$^{hi}$ cells (MoCs), SSC-A$^{lo}$CD172$^{int}$ cells, CD13$^+$ (cDC1 dendritic cells), NKp46$^+$ (NK-cells), CD21$^+$ (B-cells), CD3$^+$ T-cells, CD3$^+$CD4$^+$ T-cells, CD3$^+$CD8$^+$ T-cells are reported (4 pigs per group). Each pig is labelled with a unique symbol throughout the paper. The gating strategy is presented in S4 Fig.

We performed a similar analysis on the CFSE$^-$ MoCs. CFSE$^-$ MoCs originate from the donor and possibly from some CFSE$^-$ recruited cells. On CFSE$^-$ MoCs, the upregulation of MHC class II and of CD80/86 was non-statistically significant between 6 h and 10 h (Fig 5, middle panels, and S5 Fig), although some non-statistically significant upregulation related to inter-animal variability was observed. This finding indicates that donor MoCs do not consistently upregulate the antigen presentation molecules, differently from the recruited CFSE$^+$ MoCs. In the AMs (SSC$^{hi}$CD172A$^{hi}$CD163$^{hi}$, S6 Fig) all express MHC class II and CD80/86, therefore geometric mean intensities were used for illustration and their expression was not modified by cross-circulation (Fig 5, right panel).

Finally, we checked that the upregulation of the MHC class II and CD80/86 molecules on the perfusing pig SSC$^{lo}$CD172A$^{hi}$ MoCs was well induced by the cross-circulation process in the lung and not by a non-specific effect of the non-biological surfaces of the circuit. Indeed, in a pig blood connected to the circuit without lung, expression of MHC class II and CD80/86 was not increased, neither on CSFE$^+$ and nor on CFSE$^-$ MoCs, at 6 and 10 h (S7 Fig).

Overall, these data show that perfusion in lung grafts induces a significant upregulation of antigen presentation molecules on the recruited MoCs and not on donor MoCs and AMs.

## Discussion

We have shown here that the cross-circulatory platform using pig lung *ex vivo* is a potent method for dissecting the cellular recruitment and response to the initial encounter between donor and host in the lung, during extended periods of time, in the pertinent pig preclinical model of LT. In particular we have shown that myeloid cells, namely PMNs and MoCs, are the main recruited cell types. Furthermore, we found that recruited MoCs strongly and rapidly upregulated the MHC class II and CD80/86 molecules upon the reperfusion in the allogeneic lung, whereas donor MoCs and AMs did not. Thus recruited MoCs from the recipient rapidly acquire antigen-presentation molecules and therefore could play an important role in generating subsequent T- and B-cell allo-responses. The effectiveness of treatments controlling this activation could be evaluated in future studies using this cross-circulatory platform.

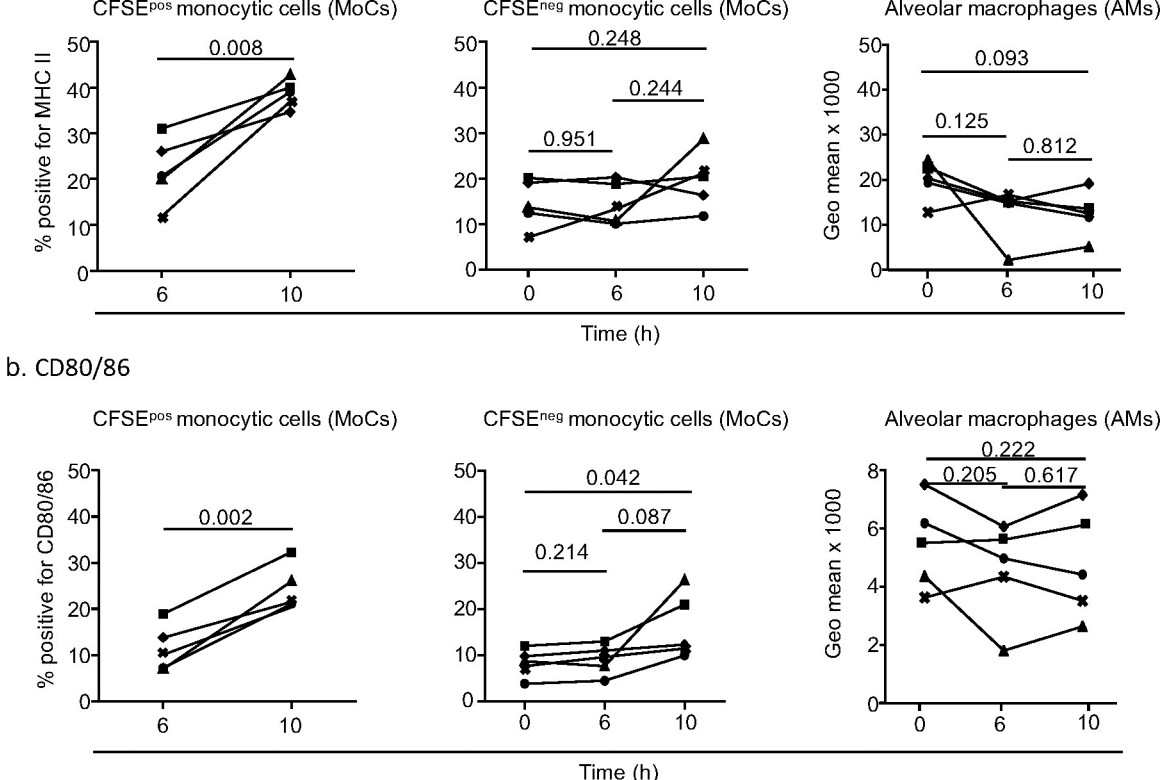

**Fig 5. Modulation of MHC class II and CD80/86 expression on lung monocytes and macrophages upon cross-circulation. a.** From left to right, percentage of MHC class II-positive cells within live CFSE⁺SSC-AᴸᵒCD172Aʰⁱ cells (CFSE⁺ monocytic cells, MoCs), within live CFSE⁻SSC-AᴸᵒCD172Aʰⁱ cells (CFSE⁻ MoCs) and expression of MHC class II (geometric mean intensity) on live alveolar macrophages (AMs, which are all positive for MHC class II) analyzed as presented in S5 and S6 Figs. **b.** Same as in **a.** for analysis of CD80/86 expression. In a and b, each pig is labelled with a unique symbol throughout the paper, five pigs per group were analyzed. After passing a normal distribution test, the values were compared between different timings with a paired t-test and the p-values are reported.

The pig preclinical model is commonly used in LT for different reasons: i) pigs and humans share a similar lung anatomy (a similar number of bronchial generations, histology, size) [25], ii) they share a similar physiology (digestion, nictemeral rhythm) and iii) the porcine immune system resembles the human one for more than 80% of analyzed parameters in contrast to the mouse that shares about 10% [10]. Therefore the pig stands as an important complementary animal model to the rodent models; whereas the latter offer incomparable advantages with targeted gene mutants for mechanistic studies, they also present anatomic limitations (size and specificities) as well as immunological properties leading to much easier establishment of tolerance to allografts and lower requirements for immunosuppressive drugs as compared to humans, therefore another translational model such as the pig is highly valuable [26]. The first surgical technique of LT was established in the pig model as well as improvements of the surgical practice and evaluation of immune suppressive therapies [26], therefore the cross-circulatory platform is highly pertinent for translational research in LT. Furthermore the cross-circulatory platform in pigs appears as a robust method due to the experimental controllability of the system. The tight regulation of the vascular flow during the process and the controlled ventilation probably explains the high degree of reproducibility regarding cell recruitment and activation parameters across experiments. Furthermore, it offers a direct and continuous accessibility to the perfused lung for

regular sampling. It is also simple and fast to perform as compared to a complete LT, with little or no suffering of the perfusing animal that could be re-used for other purposes after the experiment, provided that the exposure to the allogeneic lung would not generate biases. All these qualities and advantages would not be obtained with complete LT in rodent or pig models that induce high surgical stress, hemodynamic constraints, and require high levels of technicity and man-power support, especially in case of pig LT. Cross-circulation has been initially demonstrated to repair damaged lung tissue and maintain extracorporeal lungs healthy for extended periods of time up to several days by a team at Columbia university, New York, USA [11, 13, 15]. Therefore the use of this platform to study the first donor/recipient encounter could be extended to much longer durations. The settings of the Columbia team differed from ours regarding pig strain, donor death (beating heart vs circulatory death in our case), heating method of the circuit, anesthetic regimen (halogenated volatile versus propofol maintenance in our case) and blood reservoir. We checked that the circuit and blood reservoir had no effect on the antigen-presenting molecule expression on the perfusing pig MoCs (S7 Fig). The Columbia team also systematically treated the recipient pig with corticosteroids, which we did not do, and in the absence of corticosteroids, we found that the lung respiratory function and structures, global hematological parameters and cell viability were not affected, at least for 10 h.

The cross-circulatory platform coupled to cell mapping permitted us to show that many cell types were recruited to the allogeneic lung, mainly PMNs and MoCs and also NK cells, T-cells, B-cells and cDC1. We proceeded to cell labeling exclusively on freshly isolated cells in order to avoid bias related to cell freezing. Therefore our antibody panel was adjusted for reasons of feasibility and for instance, it did not include the complex mAb combination for cDC2 detection [18]. Importantly, the single cell RNA-seq technique could be particularly suitable to further study the cell recruitment and activation in the first encounter between donor and recipient cells. Indeed, as we used female perfusing pigs and male donors, the donor origin can be deduced from genes expressed by the Y chromosome. Using scRNA-seq, discrete cell types could be identified and well characterized with much larger signatures that what can be done with classical cytometry. In addition modulated functions and pathways in the recruited cells and donor cells could be inferred from differentially expressed genes, thereby providing a powerful system to finely investigate ischemia-reperfusion response in the allogeneic context not only in immune cell types but also in the epithelial, fibroblastic, lymphatic and endothelial cells of the graft. As cross-circulation was shown to be amenable to several days using conscious pigs [13], the innate allo-response could also be studied with scRNA-seq for a longer period than the 10 h duration used here (a time limit due to the progressive extinction of the CFSE staining, see S3 Fig). Interestingly inbred mini-pig lines could be used to compare the innate cellular responses that are induced by ischemia-reperfusion only (syngeneic model), and by allogenicity (allogeneic model). Furthermore the cellular effects of discordant MHC class I and/or MHC class II molecules and/or AO blood types between the donor and perfusing pigs could be evaluated with this cross-circulatory platform.

Here we particularly focused on the induction of the expression of MHC class II and costimulatory CD80/86 on monocytes and macrophages, given the place of these cell types in the initiation of acute and chronic rejection of allografts [24, 27]. This activation of recruited MoCs may lead to their differentiation in MPs and/or Mo-derived DCs, becoming potent antigen-presenting cells (APCs) for initiation of the T- and B-cell allo-responses [28]. Therefore this cross-circulation platform could be used to test the effectiveness of induction drugs -which are administered at the surgical step- on this recruited monocyte activation. Indeed these drugs which have not been designed to target innate immunity, include corticosteroids, calcineurin inhibitors, and/or mycophenolate, and their effects on mononuclear cells are not known and could be investigated with this platform [29]. Indeed a recent publication

demonstrated in mice that targeting a calcineurin inhibitor to myeloid phagocytes induced better protection against skin graft rejection than the conventional administration that aims at suppressing lymphocytes [30]. New therapeutic avenues such as administration of mesenchymal stem cells [31] or photopheresis [32] could be similarly tested.

Finally, cross-circulation could be used to monitor the cellular effects of surgical practices. As said above, at the difference with LT, ventilation and perfusion are highly controllable upon cross-circulation; therefore the detrimental effects on the initial donor/host cell response of forced ventilation as ventilator-induced lung injury [33] and high flow reperfusion rate upon de-clamping [34] could be analyzed in that model and thereby could lead to adjustments of practices. In addition the impact of lung procurement modalities on the cellular responses could be investigated, such as donation by circulatory death versus by brain death, as well as prolonged warm ischemia. In this paper, we used procurement upon circulatory death, that might have some effects on the type and intensity of the cellular responses that we obtained.

Overall we report on an original, relatively easy, rapid and controllable in vivo technique to study the response to the first encounter between donor lung cells and host immune cells in the pertinent pig model for LT. We show the robustness of the technique and its suitability to monitor innate cell response in an allogeneic context over time. It opens the way to a use for evaluation of anti-inflammatory and immunomodulatory regimens and testing of surgery practice adjustment, with the ultimate objective to improve LT outcome and decrease the burden of long-term immunosuppressive treatments.

## Supporting information

**S1 Fig. Introduction of the double lumen canula 20F by the Seldinger percutaneous technique for connecting the perfusing pig circulation to the extracorporeal circuit. a**. Sus sternal punction. **b.** Wire position in the inferior vena cava (white arrow) visualized by ultrasound. **c**. Catheterization with double lumen canula. **d**. View of the double lumen canula in our cross-circulation set-up.
(PDF)

**S2 Fig. Gross appearance of the lung upon cross-circulation and general view of the model. a**. Lung before cross circulation initiation. **b**. Lung after 10 h of cross-circulation. **c**. General view of our cross-circulation set-up.
(PDF)

**S3 Fig. Detection of CFSE staining stability in blood cells.** Blood was collected 1 h before cross-circulation initiation and 1 h and 8 h post cross-circulation initiation. CFSE was injected 30 min. before cross-circulation initiation. Whole blood cells were analyzed by flow cytometry after lysis of erythrocytes. The FACS profiles of CFSE fluorescence at the different timing are superimposed onto control fluorescence obtained before cross-circulation initiation and shown as a blue histogram. The percentage of CFSE[pos] cells is shown.
(PDF)

**S4 Fig. Gating strategy for the pig lung immune cell analyzes.** After gating on singlet and live cells, the workflow A was followed for CFSE[neg] cell analyses and the workflow B was followed for CFSE[pos] cell analyses. Except for PMNs, cells were analyzed using an intermediate gate on SSC-A[lo] cells to avoid noises from PMNs. PMNs, CD172A[hi] (monocytic cells, MoCs), CD172A[int], CD13[pos] (cDC1 dendritic cell subset), NKp46[pos] (NK-cells), CD21[pos](B-cells), CD3[+] T-cells, CD3[pos]CD4[pos] T-cells, CD3[pos]CD8[pos] T-cell subsets are shown.
(PDF)

**S5 Fig. MHC class II and CD80/86 expression analysis on CFSEneg (a) and CFSEpos (b) monocytic cells during cross-circulation.** Pig lung cells from cross-circulation experiments were analyzed for MHC class II and CD80/CD86 expression on monocytic cells at the indicated timing (0, 6, 10 h) on a representative pig shown in Fig 5 (animal represented as a "filled circle"). CFSE$^{pos}$ and CFSE$^{neg}$ monocytic cells (live SSC-A$^{lo}$CD172A$^{hi}$ cells) were selected as shown in Supplement 6. An IgG2a isotype control (ISC) was done on a pool of lung cells from the 0, 6 and 10 h biopsies (see material and methods). The percentage of positive cells among monocytic cells is depicted.
(PDF)

**S6 Fig. Expression of MHC class II on alveolar macrophages (AMs). a.** Upon exclusion of granulocytes on total live lung cells, AMs were identified as CD172A$^{hi}$CD163$^{hi}$ cells that back-gated on CD172A$^{hi}$SSC-A$^{hi}$ cells. Upon sorting by flow cytometry, they were stained with MGG. **b.** Expression of MHC class II and CD80/86 was analyzed for their geometric mean intensity. The control staining with a IgG2a ISC is shown.
(PDF)

**S7 Fig. MHC class II and CD80/86 expression analysis on CFSEneg (a) and CFSEpos (b) monocytic cells in a control experiment without donor lung.** In order to check for a possible activation effect of the non-biological surfaces of the circuit, PBMCs from a "mock" cross-circulation experiment without lung were analyzed for MHC class II and CD80/CD86 expression on monocytic cells at the indicated timing (0, 6, 10 h). CFSE$^{pos}$ and CFSE$^{neg}$ monocytic cells (live SSC-A$^{lo}$CD172A$^{hi}$ cells) were selected as was done for the lung cells in S4 Fig. An IgG2a isotype control (ISC) was done on a pool of cells from the 0, 6 and 10 h timing. The percentage of positive cells among monocytic cells is depicted.
(PDF)

**S1 Table. Antibodies (primary mAbs and secondary Abs) used in the study.**
(DOCX)

**S2 Table. Vital and biological parameters in perfusing pigs throughout 10 h of cross-circulation support.** The monitored parameters are reported, 5 pigs/group for vital parameters and 4 pigs/group for blood cell counts. Values represent means ± standard deviations. A Wilcoxon matched pairs signed rank test was done between initial stabilized values for the heart, respiratory, temperature and lactate parameters (at 1 h or 2 h) and 10 h and between 0 h and 10 h for all other parameters, and showed no statistically-significant difference. BP, blood pressure.
(DOCX)

# Acknowledgments

The work has benefited from the facilities and expertise of @BRIDGe (GABI, INRA-AgroParisTech, Paris-Saclay University, France) for the histology slide preparations and the scanner usage with the valuable assistance of Marthe Vilotte. We warmly thank the "Installation expérimentale porcine" of the GABI unit and in particular Pascal Lafaux and Giorgia Egidy as well as the Pig Physiology and Phenotyping Experimental Facility (https://doi.org/10.15454/1.5573932732039927E12) in particular Nelly Muller and Eloïse Delamaire. We thank Justine Cohen of the Foch Hospital for anatomopathological assessments. We are grateful to Sebastien Jacqmin, Frederic Harvengt and Nicolas Lavole for their help in managing pig perfusions. The surgery was done thanks to the Surgery platform facility CIMA, DOI: MIMA2, INRAE, 2018. Microscopy and Imaging Facility for Microbes, Animals and Foods, https://doi.org/10.15454/1.5572348210007727E12.

## Author Contributions

**Conceptualization:** Matthieu Glorion, Isabelle Schwartz-Cornil, Edouard Sage.

**Data curation:** Florentina Pascale, Isabelle Schwartz-Cornil.

**Formal analysis:** Matthieu Glorion, Florentina Pascale, Isabelle Schwartz-Cornil.

**Funding acquisition:** Matthieu Glorion, Edouard Sage.

**Investigation:** Matthieu Glorion, Florentina Pascale, Jérôme Estephan, Maxime Huriet, Carla Gouin, Céline Urien, Fany Blanc, Julie Rivière, Christophe Richard, Valérie Gelin, Julien De Wolf, Morgan Le Guen, Isabelle Schwartz-Cornil, Edouard Sage.

**Methodology:** Matthieu Glorion, Florentina Pascale, Jérôme Estephan, Maxime Huriet, Carla Gouin, Céline Urien, Fany Blanc, Julie Rivière, Christophe Richard, Valérie Gelin, Julien De Wolf, Morgan Le Guen, Antoine Magnan, Antoine Roux, Isabelle Schwartz-Cornil, Edouard Sage.

**Project administration:** Isabelle Schwartz-Cornil, Edouard Sage.

**Supervision:** Isabelle Schwartz-Cornil, Edouard Sage.

**Validation:** Morgan Le Guen, Antoine Magnan, Antoine Roux, Isabelle Schwartz-Cornil, Edouard Sage.

**Visualization:** Florentina Pascale, Isabelle Schwartz-Cornil.

**Writing – original draft:** Matthieu Glorion, Florentina Pascale, Isabelle Schwartz-Cornil.

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
