## [Decision Letter · Decision Letter 0]

8 Mar 2023

PONE-D-23-01584A cross-circulatory platform for monitoring innate responses in lung grafts

PLOS ONE

Dear Dr. Schwartz-Cornil,

Thank you for submitting your manuscript to PLOS ONE. After careful consideration, we feel that it has merit but does not fully meet PLOS ONE’s publication criteria as it currently stands. Therefore, we invite you to submit a revised version of the manuscript that addresses the points raised during the review process.

We look forward to receiving your revised manuscript.

Kind regards,

Lourdes Chacon Alberty, M.D

Academic Editor

PLOS ONE

Journal Requirements:

   "ES received funding from ADETEC-Coeur support (ES), la « Chaire Universitaire de Transplantation Université de Versailles-Saint-Quentin en Yvelines, Hôpital Foch et Fondation Foch, AR received the Association Gregory Lemarchal and the association Vaincre la Mucoviscidose (project number RF20220503016) and ISC received funding from INRAE institutional support." 

    "Authors have no competing interest."

Reviewers' comments:

Reviewer's Responses to Questions

**Comments to the Author**

1. Is the manuscript technically sound, and do the data support the conclusions?

Reviewer #1: Partly

Reviewer #2: Yes

2. Has the statistical analysis been performed appropriately and rigorously? 

Reviewer #1: Yes

Reviewer #2: Yes

3. Have the authors made all data underlying the findings in their manuscript fully available?

Reviewer #1: Yes

Reviewer #2: Yes

4. Is the manuscript presented in an intelligible fashion and written in standard English?

Reviewer #1: No

Reviewer #2: Yes

5. Review Comments to the Author

Reviewer #1: This is a very innovative study with great potential to uncover mechanisms of ischemia reperfusion injury and reveal putative targets for intervention. This area of investigation is of great interest to the lung transplant community and has translatable potential to transplantation in general. The manuscript could be enhanced through the following ways

1. The paper would benefit significantly from a native english speaking editor. There are countless awkward statements and grammatical errors throughout the text which can easily be corrected by a native english speaker.

2. The major deficit I see with this manuscript are the lack of different conditions to compare. There is novelty in the technique, but there is a lack of scientific rigor and the lack of varying conditions diminishes confidence that the observed immune phenotypes have a biological underpinning. By showing, for instance, variable injury phenotypes in response to prolonged ischemic times, this would vastly increase the significance of this research.

3. Related to point #2 above, why did the authors chose as the sole condition: a DCD donor model with additional cold ischemia time? Arguably, a cleaner model would use a heart beating donor.

4. The focus of this paper is innate immune responses to ischemia reperfusion injury. I am confused as to why the authors go great lengths to introduce and explain that they specifically chose outbred pigs of differing genders to introduce an "allogeneic" component to the model. If the goal is to understand innate immunity, the model should use syngeneic transplants.

5. Throughout the manuscript, the word "recruited" is used liberally to describe perfusing pig immune cells found in the allograft. This word should be used carefully as "recruited" implies an active process. Passively perfusing the lung with CSFE labeled blood is not an active biologic process...therefore CSFE+ cells in the intravascular space I would not consider as recruitment. Conversely, CSFE+ cells in the alveolar space or interstitium could be considered as "recruited" since there was a process of extravasation. Related to this comment, Fig 3 could be modified to demonstrate this with the IHC showing improved histologic resolution to identify which compartment the CSFE+ cells are in. Currently, that figure is blurry and zoomed out too far so you cannot tell histologic architecture.

6. Fig 2 should include 0, 6, and 10 hr timepoints for lung histology to assess over time. For P/F ratio, is should be better defined where they are drawing samples from.

7. Fig 5, how do you define MHCII positive cells? Need to show the FACS plots.

8. In the discussion I would tone down the discourse on the fallibility of rodent models to study IRI. The mouse is a much higher throughput model of lung IRI and an extremely well established model. Multiple groups throughout time have used both hilar clamp IRI models and even lung transplant models in mice to uncover the immune mechanisms of IRI. Pigs are much more expensive to use. Also, lung transplant techniques were developed in Dogs more so than pigs.

9. A significant weakness in the described cross perfusion platform that needs to be mentioned by the authors is the extremely well known pro-inflammatory stimulus that is the cardiopulmonary bypass circuit (of which all the elements are in this model: pump, membrane oxygenator, reservoir, and LOTS of plastic tubing). Every clinician that uses mechanical circulatory support knows that the blood-plastic interface activates the complement system, coagulation system, and releases a host of pro-inflammatory mediators into circulation. This will certainly affect any study evaluating the innate immune response.

Reviewer #2: Very nice observational molecular work by the French Lung Transplant Group and Prof. Sage. In the current study they used 10 donors and 10 recipients and cross-circulation for 10 hours. The method cross-circulation itself is not novel but the molecular work in regards to cross-ciculation and recrutment of immune cells with a strong clinical translational potential to the donr lung has not been done by the Colombia team or any other group that has implemented the method as far as I know.

I have some additional questions.

Why did the authors set the time to 10 hours of circulation? To make the connection between acute and chronic rejection wouldn’t it be better and more translational to run over 4 days or even more? An unharmed lung would probably last longer on cross-circulation than the injured lungs at Colombia.

Why was a DCD lung donation model used? Would there be any differences using a beating heart donor lung? Brain dead pig models are extremely demanding to achieve but why not just use a beating heart donor?

Was the donor and the recipient matched according to blood type?

Was the lung really put on ice before establishment of cross-circulation? Perhaps it was cold storage? Either way I cannot find the ischemic time in the manuscript please add.

Did the recipient receive any immunosuppressant therapy? If not, why. Please elaborate.

The manuscript lacks many of the hemodynamic, and ventilatory parameters for the 10 hours. Would be interesting to see for example flow through the “donor lung”, oxygen, saturation, end-tidal CO2, compliance, and respiratory rate were continuously monitored. If I understood, it right this have been measured in this model. Perhaps it is in the supplementary file. I was unable to open those. If so, please move them to the main manuscript.

It is a nice manuscript with high quality molecular work. However, if any, this reviewer lacks a interventional angle to increase the novelty of the work.

6. PLOS authors have the option to publish the peer review history of their article (what does this mean?). If published, this will include your full peer review and any attached files.

Reviewer #1: No

Reviewer #2: No

---

## [Author Response · Author response to Decision Letter 0]

18 Mar 2023

Jouy en Josas, March 18th 2023

Dear Editor of PLOS ONE,

We have provided responses to the reviewers’ queries which you will find in the "Responses to reviewers" section (located at the end of the pdf file as requested by the PLOS ONE platform). The reviewer’s text is in italic, our response is in standard font, the modified text in the manuscript is in red. We hope that our responses will be satisfactory to you and the reviewers, so that the paper can be published in PLOS ONE. 

Isabelle Schwartz-Cornil, on the behalf of the authors of the manuscript PONE-D-23-01584

---

## [Decision Letter · Decision Letter 1]

2 May 2023

A cross-circulatory platform for monitoring innate allo-responses in lung grafts

PONE-D-23-01584R1

Dear Dr. Schwartz,

We’re pleased to inform you that your manuscript has been judged scientifically suitable for publication and will be formally accepted for publication once it meets all outstanding technical requirements.

Kind regards,

Lourdes Chacon Alberty, M.D

Academic Editor

PLOS ONE

Additional Editor Comments (optional):

Reviewers' comments:

Reviewer's Responses to Questions

**Comments to the Author**

1. If the authors have adequately addressed your comments raised in a previous round of review and you feel that this manuscript is now acceptable for publication, you may indicate that here to bypass the “Comments to the Author” section, enter your conflict of interest statement in the “Confidential to Editor” section, and submit your "Accept" recommendation.

Reviewer #1: All comments have been addressed

2. Is the manuscript technically sound, and do the data support the conclusions?

Reviewer #1: Yes

3. Has the statistical analysis been performed appropriately and rigorously? 

Reviewer #1: Yes

4. Have the authors made all data underlying the findings in their manuscript fully available?

Reviewer #1: Yes

5. Is the manuscript presented in an intelligible fashion and written in standard English?

Reviewer #1: No

6. Review Comments to the Author

Reviewer #1: Thank you for addressing the points. This article is a valuable addition to the literature on lung transplant, especially with the description of the technique of cross circulation

7. PLOS authors have the option to publish the peer review history of their article (what does this mean?). If published, this will include your full peer review and any attached files.

Reviewer #1: No

---

## [Editor Report · Acceptance letter]

19 May 2023

PONE-D-23-01584R1 

A cross-circulatory platform for monitoring innate allo-responses in lung grafts 

Dear Dr. Schwartz-Cornil:

I'm pleased to inform you that your manuscript has been deemed suitable for publication in PLOS ONE. Congratulations! Your manuscript is now with our production department. 

Kind regards, 

on behalf of

Dr. Lourdes Chacon Alberty 

Academic Editor

PLOS ONE